# Effect of Irradiation on Corrosion Behavior of 316L Steel in Lead-Bismuth Eutectic with Different Oxygen Concentrations

**Nariaki Okubo \*, Yuki Fujimura and Masakatsu Tomobe**

Nuclear Science and Engineering Center, Japan Atomic Energy Agency (JAEA), Tokai, Ibaraki 319-1195, Japan; fujimura.yuki@jaea.go.jp (Y.F.); tomobe.masakatsu@jaea.go.jp (M.T.)

\* Correspondence: okubo.nariaki@jaea.go.jp; Tel.: +81-29-282-6212

**Abstract:** In an accelerator-driven system (ADS), the beam window material of the spallation neutron target is heavily irradiated under severe conditions, in which the radiation damage and corrosion co-occur because of high-energy neutron and/or proton irradiation in the lead–bismuth flow. The materials used in ADSs must be compatible with the liquid metal (lead–bismuth eutectic (LBE)) to prevent issues such as liquid metal embrittlement (LME) and liquid metal corrosion (LMC). This study considers the LMC behavior after ion irradiation of 316L austenitic steel for self-ion irradiations followed by the corrosion tests in LBE with critical oxygen concentration. The 316L samples were irradiated by 10.5 MeV-$Fe^{3+}$ ions at a temperature of 450 °C, up to 50 displacements per atom (dpa). After the corrosion test performed at 450 °C in LBE with low oxygen concentration, a surface of the nonirradiated area was not oxidized but appeared with locally corrosive morphology, Ni depletion, whereas an iron/chromium oxide layer fully covered the irradiated area. In the case of the corrosion surface with high oxygen concentration in LBE, the surface of the nonirradiated area was covered by an iron oxide layer only, whereas the irradiated area was covered by the duplex layers comprising iron and iron/chromium oxides. It is suggested that irradiation can enhance the oxide layer formation because of the enhancement of Fe and/or oxygen diffusion induced by the radiation defects in 316L steel.

**Keywords:** accelerator-driven system (ADS); liquid metal corrosion (LMC); lead–bismuth eutectic (LBE); self-ion irradiation; oxygen concentration in LBE; irradiation effect on corrosion behavior

## 1. Introduction

Decreasing the risk of spent nuclear fuel elements has become a major concern, especially in Japan after Fukushima's first nuclear power plant accident. An accelerator-driven system (ADS) is an important concept to realize partitioning and transmutation [1] to reduce the hazards associated with the spent fuel. In an ADS, the beam window, which is the boundary between a high-energy accelerator for protons in a vacuum and a spallation target of lead–bismuth eutectic (LBE), is irradiated under severe conditions to sufficiently transmute the minor actinides in the fuel cladding. The ADS irradiation conditions, which induce considerable displacement damage with high concentrations of helium (He) and hydrogen (H) atoms in the materials, are produced by the high-energy proton and spallation neutron irradiation. Degradation of the mechanical and corrosion properties after irradiation in the LBE at system temperatures, e.g., from 350 to 550 °C, should be maintained within a permissible range for a good system design [2]. High-fluence neutron irradiation experiments with up to about 20 and 100 displacements per atom (dpa, the parameter for indicating the radiation damage level) are the estimated upper limits of the irradiation damage tolerated by the beam window and cladding materials in ADS [2], respectively; however, these are practically challenging to execute using an experimental nuclear reactor because of the required long irradiation time.

An oxygen control for the LBE coolant is critical in ADSs, and the control system needs to be properly installed in the LBE coolant system. The oxygen concentration of LBE

must be controlled below the upper concentration of lead oxide (PbO) formation because PbO results in stuck valves and plugs the narrow flow channels of LBE during long-term operation. In steel materials, the surface-protective layer of iron and/or chromium oxide dissolves when the oxygen concentration is low, reducing the metal oxide. The acceptable oxygen concentration depends on the LBE temperature and ranges from $10^{-5}$ to $10^{-7}$ wt.% at 450 °C [3]. These upper and lower oxygen concentrations are critical values of the oxygen concentration boundary and need to be controlled in the system for the LBE in ADS.

The material corrosion behavior in LBE has been studied for approximately two decades [4,5], and the corrosion kinetics of steels, such as austenitic steels, 316L, and ferritic/martensitic steels, T91, has been discussed [6–8]. In the case of 316L steels, it is well-known that Ni, which is one of the main austenite formers, can dissolve in the LBE, because of the high solubility in LBE. The depleted area of Ni appears to be ferritic phase [9], followed by the substitutive Pb and/or Bi penetration [10], especially at a higher temperature of approximately 500 °C, under flowing LBE [5] and long-time corrosion test over 3000 h, even at a relatively low temperature, such as 450 °C [11].

The few studies [7,8] that have investigated the effect of neutron irradiation on the material properties in LBE have been conducted only from the viewpoint of liquid metal embrittlement (LME); thus, reports concerning liquid metal corrosion (LMC) of irradiated materials are limited. However, ion irradiation is a powerful technique to simulate ADS irradiation conditions with accurate temperature control, making it appropriate to investigate the mechanism of microstructural evolution under irradiation and further select candidate materials before irradiation by neutrons. Firstly, the formation of oxide layers on the surface of T91 steel was reported to have improved because of the triple ion irradiation of Fe, He, and H beams, which simulated the ADS spallation neutron irradiation even after the corrosion test conducted under the low oxygen concentration in LBE [12]. As per the report, only Fe ion irradiation for T91 specimens proved incongruous to the oxide-formation enhancement. It was suggested that the vacancy defects, which were effectively induced by the triple ion irradiations in T91, played a vital role in enhancing the oxide formation. On the contrary, displacement damage can produce vacancy defects in 316L steels [13]. In this study, the LMC behavior after irradiation is considered for self-ion irradiations without simultaneous He and H irradiations, followed by the corrosion tests to study the effect of single irradiation on the corrosion of 316L steels.

## 2. Experiments

In an ADS, SS316L (316L) is one of the candidate materials for the in-core structure. The material composition of 316L steel is shown in Table 1. The 316L steel was solution-annealed at 1040 °C for 10 min and then cooled in water. A specimen with a length of 6.0 mm, width of 3.0 mm, and thickness of 1.0 mm was cut from the bulk sample to mechanically polish and finish its surface via buffing, using 50 nm alumina nanoparticles for a mirror finish, thus achieving a final thickness of approximately 0.75 mm. As explained below, two such specimens were set in the irradiation specimen holder with the same irradiation condition but different corrosion conditions.

**Table 1.** Chemical composition of 316L steel (wt.%).

| Fe | Cr | Ni | Mo | Mn | Si | P | C | S |
|----|----|----|----|----|----|----|----|----|
| Bal. | 17.46 | 12.11 | 2.19 | 0.82 | 0.51 | 0.027 | 0.017 | 0.001 |

The 316L specimens were irradiated by 10.5 MeV-$Fe^{3+}$ ions at a temperature of 450 °C up to about 50 dpa. Ion irradiation experiments were conducted at Takasaki Ion Accelerators for Advanced Radiation Application, Japan's National Institutes for Quantum and Radiological Science and Technology (QST). As shown in Figure 1a, two specimens were held by a 10 mmφ steel mask, and the irradiation area was covered by an aluminum foil to create a nonirradiated area on the surface of the same specimen, having almost the same experimental condition as the temperature history during ion irradiation and also immer-

sion in LBE. Irradiation experimental conditions were determined using the Stopping and Range of Ions in Matter (SRIM) code [14]. The displacement damage depth profile is shown in Figure 1b. The schematic cross-sectional image of the irradiated specimen was also inserted. Both irradiation and corrosion temperature were controlled at 450 °C because the medium ADS component temperature is around 450 °C [2]. An infrared pyrometer (NIKON) monitored the surface temperature under irradiation and actively controlled it via electron bombardment heating, Joule heating, and beam heating to achieve the accurate irradiation temperature. Displacement damage in the corrosion test was determined to be approximately 4 and 8 dpa at the specimen surface. For example, the specimen was irradiated up to 8 dpa at its surface and 52 dpa at a depth of 2 μm, as shown in Figure 1b. In the case of 4 dpa, the depth profile of the displacement damage represented half of the whole depth. After the ion irradiation, the specimen was immersed at a lower temperature compared with the corrosion test temperature in the LBE pot, keeping the LBE at the desired oxygen concentration. The specimen was fixed in LBE using a stainless-steel wire connected to a tungsten weight. The experimental setup and further details of the corrosion test are provided in [12]. The oxygen concentration in LBE was measured using a Pt/air type-6YSZ (yttria-stabilized zirconia) oxygen sensor (fabricated by JAEA [15]) and maintained using a covering gas of premixed Ar + 5.0%$H_2$ for a low oxygen concentration spanning the order of $10^{-8}$–$10^{-9}$ wt.%, which was around the lower critical concentration for oxide ($Fe_3O_4$ and FeO) formation [3]. The corrosion test time started when the LBE temperature reached 450 ± 5 °C, shown in Figure 2 as an open triangle. Notably, during the initial 10% of the immersion period, approximately 35 h, high oxygen concentration in the range of $2.4 \times 10^{-4}$–$2.3 \times 10^{-8}$ wt.% was employed because of the opening of the pot to set the specimens into the LBE, as shown in Figure 2a. Consequently, the incident of this higher oxygen concentration highlighted the irradiation effect for the ion-irradiated specimen before reaching a low oxygen concentration. Before the corrosion test's completion under lower oxygen concentration, an unforeseen leakage of approximately $6.0 \times 10^{-8}$ wt.% was observed, but it could be considered insignificant compared with the oxygen concentration at working conditions. After conducting the corrosion test at the lower oxygen concentration, another test was conducted under a saturated oxygen concentration of $3.6 \times 10^{-4}$ wt.% at 450 °C, as shown in Figure 2b. In a previous study [12], the corrosion time of 1000 h was too long to observe the effect of irradiation on the specimen, especially at low oxygen concentration at approximately $10^{-8}$ and $10^{-9}$ wt.%, when comparing the radiation damage depth range of about 2 μm to the oxide thickness. Hence, we chose to operate at a third of the 1000 h, i.e., 330 h. After completing the corrosion test in 330 h, the LBE on the surface of the specimen was removed using silicone oil at 200 °C to melt the LBE into the oil bath. The silicone oil exhibited a high affinity for the corroded surface, thus penetrating the oxide layer with ease, as partially shown in the latter energy-dispersive X-ray spectroscopy (EDS) spectra. Removing the silicone oil was difficult and required care to avoid unexpected distortions of the corrosion behavior. To identify the oxide layer, X-ray diffraction (XRD; MAC Science, MXP3 with a Cu Kα X-ray source) was performed after the corrosion test. The surface observation before LBE removal and the cross-sectional corrosion behavior were assessed using a field-emission scanning electron microscope (high-resolution FE-SEM; Zeiss, Sigma) at an acceleration voltage of 30 kV. Two types of SEM detector modes of normal secondary electron and AsB (angular selective backscattering electron, showing material contrast and topographical information) were used for cross-sectional observation after fixing in electrically conductive resin via hot pressing, polishing to a mirror surface, and depositing a thin Os coating (thickness of a few nanometers) to avoid charging up under SEM observations. EDS was also performed to obtain line analytical data and mapping images of material elements in the specimens after corrosion.

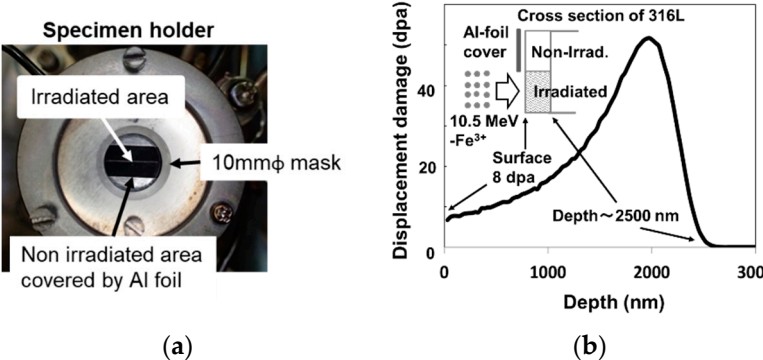

**Figure 1.** (**a**) Irradiation specimen holder and (**b**) depth distribution of displacement damage of ion irradiation. The experimental dpa was determined based on the depth distribution.

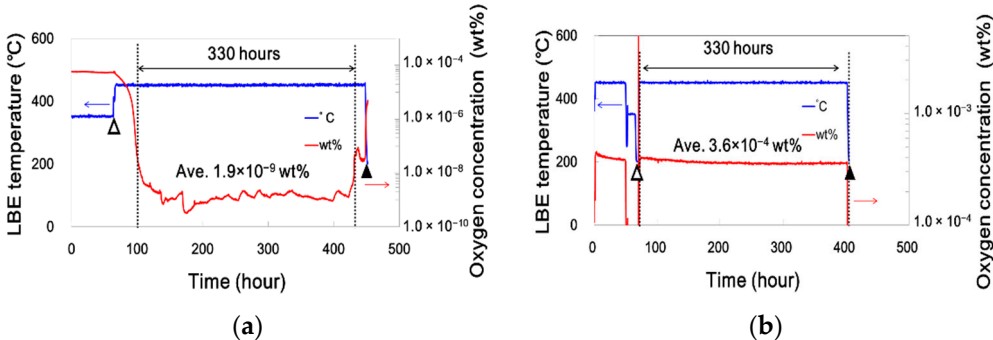

**Figure 2.** LBE temperature and oxygen concentration under corrosion tests for the 316L specimens. The corrosion tests were conducted for 330 h at (**a**) low oxygen concentration and (**b**) high (saturated) oxygen concentration in LBE at 450 °C. The triangles denote the insertion and extraction moments of the specimens as the open and solid triangles, respectively.

## 3. Results

After the corrosion test at 450 °C in LBE with a sufficiently low oxygen concentration, a surface within the nonirradiated area was left unoxidized, but a locally corrosive morphology was observed; however, an iron/chromium oxide layer covered the irradiated area. Ion irradiation of 10.5 MeV-$Fe^{3+}$ up to 8 dpa at the specimen surface was conducted at 450 °C, followed by a corrosion test in LBE at 450 °C for 330 h under the low oxygen concentration (in the range of $10^{-8}$–$10^{-9}$ wt.%, as shown in Figure 2a). Figure 3 shows typical surface SEM images observed on the boundary area between the nonirradiated (masked) and irradiated areas after removing LBE. The irradiation displacement damage was 4 and 8 dpa for the specimen shown in Figure 3a,b, respectively. The boundary lines between nonirradiated and irradiated areas clearly appeared in both upper images. In the 4 dpa irradiation, the irradiated surface showed a patchy pattern texture, which is nonuniform, whereas the covered surface showed a uniform texture in the 8 dpa irradiation. These surface SEM images were observed before cleaning the LBE with silicone oil. The lower images are a high-magnification version of both the nonirradiated and irradiated areas. As shown in the bottom left images of Figure 3a,b, the nonirradiated areas of both specimens exhibited similar local corrosion, observed as black dots. The area irradiated to 8 dpa became fully covered by the rough texture, as it was in the case of 4 dpa, indicated as a round shape with white contrast on the upper part of Figure 3a. The white contrast textures in the Figure 3a upper section comprised the elements of iron, chromium, and oxygen, as confirmed by EDS measurements (not shown in this report). The uniform texture in Figure 3b seen in the upper image comprises iron, chromium, nickel, oxygen, lead, and bismuth. Comparing the surface images among nonirradiated, irradiated, 4 dpa, and 8 dpa irradiated areas suggested that the irradiation affects the corrosion behavior of 316L in LBE post-ion irradiation. Figure 4 shows the cross-sectional SEM images of

nonirradiated and irradiated areas on the surface after a corrosion test conducted at 450 °C for 330 h. These cross-sectional SEM images correspond to the surface SEM images of identical specimens shown in Figure 3, but the surface LBE was removed using silicone oil. The upper and lower images denote: (a), (c) nonirradiated area and (b), (d) irradiated area of 4 and 8 dpa specimens, respectively. The oxygen concentration was low in LBE at 450 °C. In the cases of nonirradiated sites for both specimens, local corrosion was observed on both surfaces as a weak contrast revealed by the AsB detector, which could show the compositional difference, as indicated in Figure 4a,c. The local corrosion size was estimated to be approximately 100–300 nm in diameter and about 100–150 nm in depth, measured to a higher magnification via high-resolution FE-SEM. This local corrosion observed in the cross-sectional image corresponds to the black dots on the surface, as shown in Figure 3. The Ni component of steel is considered to have dissolved in the LBE at low oxygen concentration because the oxygen concentration is close to or lower than that necessary for a Fe and/or Cr surface oxide formation. Although the nonirradiated area appeared to have almost no oxide formation, rather local corrosion, an oxide layer of about 140 nm thickness was formed on the 4 dpa irradiated area by efficiently reacting with the small amount of oxygen present within the initial 10% immersion period, as shown in Figure 2a. The oxide area occupied approximately 70% of the total irradiated area in the case of 4 dpa, and some other residual areas appeared to onset the ferritization via Ni-depletion. Unfortunately, the EDS line spectra for all the images in Figure 4 did not exhibit the distinct reduction of Ni concentration. It is obvious to consider the initial state of ferritization by Ni-depletion, as depicted in Figure 4. However, in the 8 dpa, fully irradiated areas were covered by an approximately 230 mm thick oxide layer. The irradiation enhanced the oxide formation even though the oxygen concentration was relatively low.

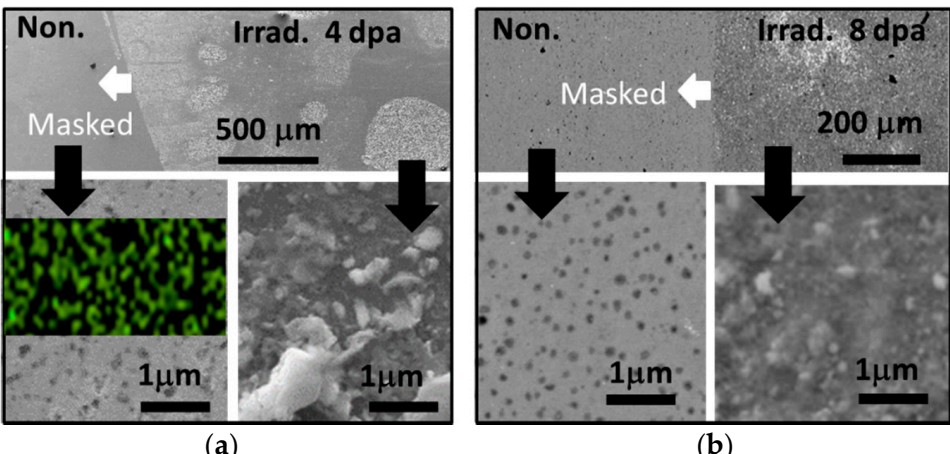

**Figure 3.** Surface SEM images after corrosion tests for the irradiated 316L specimens. Upper images were observed on the boundary area irradiated at (**a**) 4 dpa and (**b**) 8 dpa. The mapping image from Ni is inserted in the masked area shown in (**a**). The black dots correspond to the Ni-depleted areas and behave similarly for both (**a**,**b**).

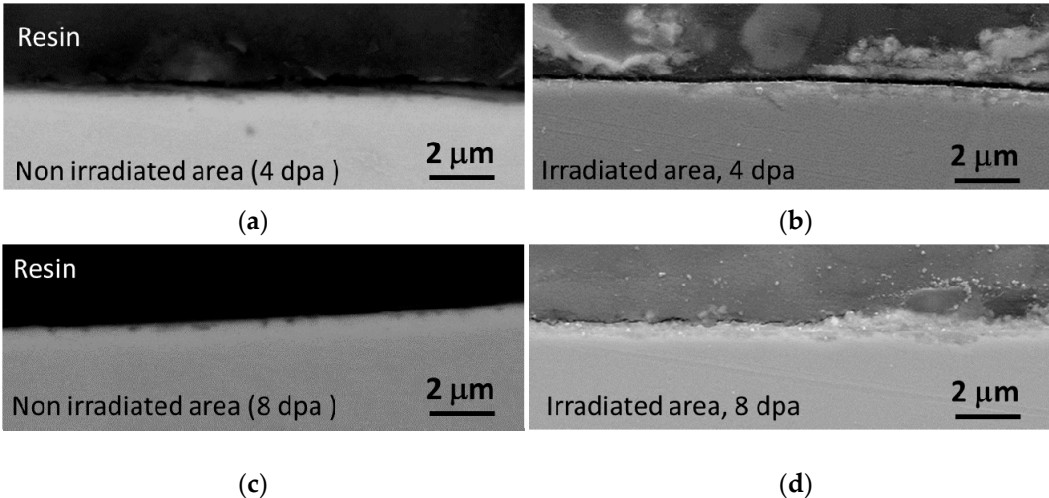

**Figure 4.** Cross-sectional SEM images after conducting corrosion tests for the irradiated 316L specimens. Upper and lower images denote: (**a**,**c**) nonirradiated area and (**b**,**d**) irradiated area of specimens irradiated at 4 and 8 dpa, respectively. The oxygen concentration was in the order of $10^{-9}$ wt.% in LBE. The images (**a**,**c**) and (**b**,**d**) were observed by AsB and SE detectors respectively, for convenience.

In the case of high oxygen concentration in LBE, the surface of the nonirradiated area reacted to form an iron oxide layer, whereas the irradiated area developed duplex layers of iron and iron/chromium oxides as coverings. A corrosion test for the other pieces of the irradiated specimens was conducted for 330 h in LBE at a saturated oxygen concentration of $2.4 \times 10^{-4}$ wt.%. SEM images and EDS line spectra of the nonirradiated and irradiated areas are shown in Figure 5. The EDS spectra from the resin included Si and O signals from silicone oil and Fe. In contrast, the other elements of steel came from the residues in the oil because the acetone cleaning was insufficient to clean the silicone oil of its residues. Even though both areas were subjected to the same preparation process, the morphologies of the surface in the two regions were different. The surface of the nonirradiated area was rough and appeared to have a thin white contrast, whereas that of the irradiated area was coarse and contained numerous cracks. The EDS line spectra in the regions are also shown in Figure 5b,c. An oxide layer with a thickness of half a micrometer, observed as a thin white layer, was formed on the surface of the nonirradiated region. In contrast, a distinct oxide layer with a thickness of approximately 1 μm, which is twice that in the nonirradiated area, covered the surface of the irradiated area. The oxygen diffused into the 1 μm depth from the original surface, which was considered to be approximately the half depth of two dotted lines, as shown in Figure 5c. The breakage of the oxide layer was caused by polishing in the irradiated region. XRD analysis revealed that the oxide layers of the nonirradiated and irradiated areas mainly comprised $Fe_3O_4$ (magnetite) and small signals of $FeCr_2O_4$ (spinel). The boundary is not very clear for the irradiated area, but it is somewhat distinguishable, as shown in Figure 5c, upper SEM image. In the case of 316L steel after immersion in LBE under a saturated oxygen concentration of $2.4 \times 10^{-4}$ wt.%, the surface of the irradiated area was coated with duplex oxide layers, as shown in Figure 5c. For corrosion in LBE with a sufficiently high oxygen concentration (in the range of $10^{-4}$–$10^{-6}$ wt.%) and/or a long corrosion time (typically over 1000 h), a duplex oxide layer forms on the steel surface [5,16]. In this duplex, the outer layer is $Fe_3O_4$ and the inner layer is $FeCr_2O_4$. In this study, the duplex layer could not be observed in the SEM images of the irradiated area in Figure 4 and the nonirradiated site in Figure 5b because the oxygen concentration was too low and corrosion time too short for the formation of the duplex layer. Nevertheless, ion irradiation up to 8 dpa enhanced the oxidation reaction by effectively using the oxygen introduced into LBE at the beginning of the corrosion test, as shown in Figure 2a, even at a low oxygen concentration, as shown in Figure 4d. The original surface is assumed to be located between

the outer and inner layers of the duplex, indicating that the inner layer of the irradiated region includes radiation damage.

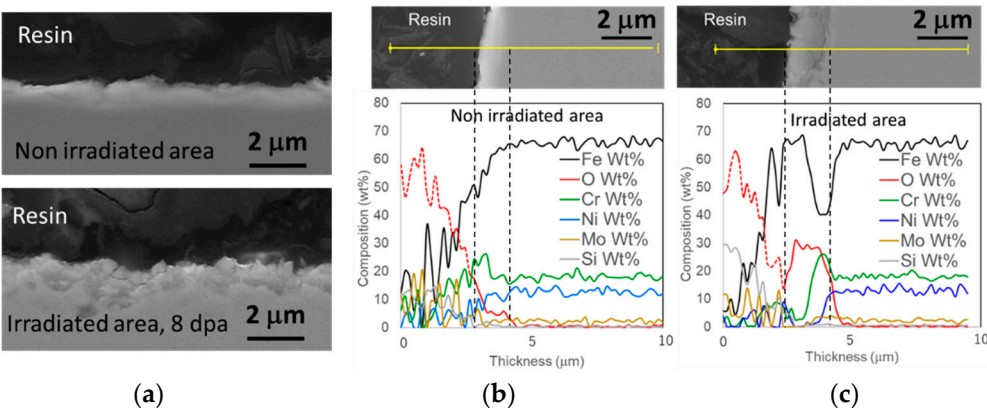

(**a**)          (**b**)          (**c**)

**Figure 5.** Cross-sectional SEM images and EDS line spectra after conducting corrosion tests for the specimen irradiated up to 8 dpa. Upper and lower images of (**a**) correspond to that of nonirradiated and irradiated areas, respectively. Figures (**b**,**c**) denote the compositions of nonirradiated and irradiated areas respectively, measured by an EDS line analysis, which was shown in the upper image. The oxygen concentration was $2.4 \times 10^{-4}$ wt.% in LBE at 450 °C. The red dotted and gray lines are signals come from the compositions of silicone oil.

In 316L steel, after conducting the corrosion test in LBE, a cross-sectional SEM image captured at a region exposed to ion irradiation showed a thicker oxide layer than the nonirradiated surface, even at a low oxygen concentration in LBE. The mean thicknesses of the local corrosion (depletion of Ni) oxide layers in nonirradiated and irradiated areas are shown in Figure 6. The total thickness of the duplex layer was also measured from each image in the case of the saturated oxygen concentration. In the case of the low oxygen concentration of an order of $10^{-9}$ wt.%, the irradiation enhances the oxide formation by a factor of two at maximum, and the irradiation effect on the oxidization increases with increasing displacement damage up to 8 dpa. In the case of a saturated oxygen concentration, the irradiation enhances the oxide formation by a factor of two, and the irradiation is undoubtedly effective in the surface oxidization; however, it appears to increase the displacement damage up to 8 dpa.

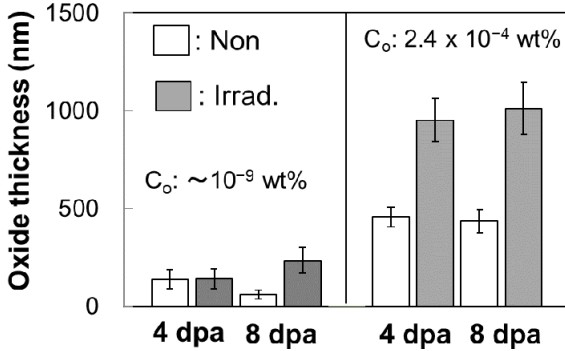

**Figure 6.** Oxide thickness of non-irradiated and irradiated areas after corrosion tests at low and saturated oxygen concentrations for 316L specimens irradiated at 4 and 8 dpa.

## 4. Discussion

The process of inward diffusion of oxygen from the LBE side and outward diffusion of Fe from the steel matrix mutually governed the oxide formation at the boundary between the steel surface and LBE. The cold working is a good simulation of the microstructural evolution under radiation damage in dislocation defects and interstitial atom types of

defects. In the case of the corrosion behavior exhibited by the cold-worked austenitic stainless-steels in LBE, the cold working accelerated the formation of the duplex oxide layer and the ferritization via Ni dissolution [10] at 500–550 °C in 1000–3000 h. As shown in Figure 5c, even at 450 °C and in 330 h experimental conditions, Ni dissolution was observed in the irradiated region and not in the nonirradiated region. As shown in Figure 6, oxide formation was enhanced via ion irradiation in LBE even at a low oxygen concentration by the formation of radiation defects. In the nonirradiated region, the oxide layer can grow in LBE with an oxygen concentration dissolved in LBE under thermal equilibrium. The thermal equilibrium state is also applicable to the irradiated region because oxidation reactions occur after ion irradiation. However, the number of Fe atoms diffusing outward might increase through site changes with vacancies, which might be trapped by impurities or the strain field induced via ion irradiation, leading to more Fe self-diffusion than that found under normal thermal activity. The activation energy of Fe self-diffusion in $\alpha$-Fe is 2.87 eV, in which the vacancy formation energy ($E_{vf}$) and migration energy ($E_{vm}$) are 1.61 and 1.3 eV, respectively [17]. The diffusion coefficient (D) is calculated by $D = D_0 \exp(-(E_{vf} + E_{vm})/kT$, where $D_0$ is $8.0 \times 10^{-5}$ m$^2$/s, k is $8.62 \times 10^{-5}$ eV/K, and T is 723 K. The thermal diffusion length ($L_d$) is described by $L_d = (6Dt)^{1/2}$, and $L_d$ of Fe was calculated to be 1.4 nm for the specimen subjected to the corrosion test performed at 450 °C for 330 h. Therefore, Fe self-diffusion does not affect oxide formation at the specimen surface. However, after ion irradiation, assuming all vacancies exist even at RT, when $E_{vf} = 0$, which means the neighbor of the diffusive atom is a vacancy, D exhibits its maximum value and $L_d$ is 580 µm. The actual $L_d$ is smaller than this maximum value because of the void formation and residual vacancies trapped by the strain field induced by radiation damage. From an SRIM calculation shown in Figure 1b, the total number of vacancies was estimated to be $1.5 \times 10^{19}$ m$^{-2}$ around 1 µm depth, induced by the displacement damage of 20 dpa, which is considered average damage through 2 µm depth. This is just the case of $E_{vf} = 0$, then, $L_d$ shows a maximum value of 580 µm.

On the contrary, using the void size of 1.3 nm and the density of $3 \times 10^{24}$ m$^{-3}$ for 316L irradiated by 4 MeV-Au ions at 450 °C up to about 20 dpa from Reference [13], the number of vacancies included in the total voids was estimated to be $4.0 \times 10^{17}$ m$^{-2}$, assuming that the one void had 100 vacancies. From these rough estimations, the number ratio of residual (invisible) vacancies is 0.0267 ($4.0 \times 10^{17}/1.5 \times 10^{19}$). Then, a mean free path (MFP) of vacancy diffusion is approximately 15 µm. This value is reasonable because recombining with interstitials and/or disappearance into sink sites reduces the MFP to be several micrometers. However, this might be a limitation of the simulation method that employed (ex situ) the corrosion test after ion irradiation.

In contrast, oxygen atoms diffuse inward from the surface as interstitial atoms. The migration energy of an interstitial atom ($E_{im}$) is 0.89 eV [14]. Here, $D = D_0 \exp(-E_{im})/kT$, where $D_0$ is $1.79 \times 10^{-7}$ m$^2$/s, k is $8.62 \times 10^{-5}$ eV/K, and T is 723 K. The oxygen diffusion length ($L_d$) was calculated to be 910 µm according to $L_d = (6Dt)1/2$. Therefore, oxygen can sufficiently diffuse into the material in this corrosion experiment in LBE. Comparing the EDS line spectra shown in Figure 5b,c, the inward oxygen diffusion lengths of the nonirradiated and irradiated regions are comparable. However, in the irradiation region, the oxygen concentration increased about six times more than that of the nonirradiated case. From these rough estimations, two reasons for the enhanced oxide formation in the case of ion irradiation in 316L are considered: (1) enhanced Fe diffusion caused by vacancy diffusion after ion irradiation and (2) enhancement of O interstitial diffusion induced via radiation damage. This enhances the oxidation reaction between Fe, Cr, and O. Based on the results of this study, it is suggested that radiation-induced diffusion during irradiation enhances the oxidation much more than that after irradiation.

## 5. Conclusions

After ion irradiation of 316L austenitic steel, the LMC behavior was studied for self-ion irradiations, followed by the corrosion tests in LBE with different oxygen concentrations.

The 316L specimens were irradiated by 10.5 MeV-$Fe^{3+}$ ions at a temperature of 450 °C, up to 8 dpa at the surface. After conducting the corrosion test at 450 °C in LBE with low oxygen concentration, a surface of the nonirradiated area was not oxidized, but corrosive morphology appeared, whereas an iron/chromium oxide layer covered the irradiated area. In the case of a high oxygen concentration in LBE, the surface of the nonirradiated area was oxidized to an iron oxide layer, whereas the irradiated area was covered by the duplex layers of iron and iron/chromium oxides. It is suggested that irradiation can advance the oxide layer formation because of the enhancement of Fe and/or oxygen diffusion induced by the radiation defects in the 316L steel.

**Author Contributions:** Conceptualization, N.O.; SEM observation, Y.F. and M.T.; formal analysis, N.O.; investigation, N.O.; writing—original draft preparation, N.O. All authors have read and agreed to the published version of the manuscript.

**Funding:** This research received no external funding.

**Acknowledgments:** These ion irradiation experiments were performed under the Shared Use Program of QST Facilities. The authors are grateful to S. Ukai for useful comments.

**Conflicts of Interest:** The authors declare no conflict of interest.

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
