# Peer review of "Effect of Irradiation on Corrosion Behavior of 316L Steel in Lead-Bismuth Eutectic with Different Oxygen Concentrations"

_qubs, doi:10.3390/qubs5030027_

Round 1
Reviewer 1 Report
Dear authors,
congratulations for this quite interesting study on the effect of irradiation on the liquid metal corrosion (LMC) behavior of 316L austenitic stainless steels in contact with liquid LBE. Your study confirms prior assumptions on the possible effect of irradiation on the LMC behavior of such steels and is in agreement with similar observations regarding the performance of stainless steels in contact with molten salts, another highly corrosive medium that is of relevance for molten salt-cooled fission reactors.
Some recommendations for further improvement of your manuscript:
- The English must be improved. Even though the quality of written English is acceptable, it needs perfection by a native English speaker to meet the standards of the Journal and to also do justice to your very nice work.
- The description of the irradiation experiment is a bit unclear. The way I understand it is: you irradiated two 316L specimens (each: 6 x 3 x 0.7 mm); one of them to 4 dpa and the other to 8 dpa (on the surface). This means that you removed the 1st specimen out of the irradiation holder after it has reached 4 dpa, is that correct? Pls clarify in the text. Moreover, the "mask" that you used to retain part of the specimen surface in the non-irradiated state: was it placed in the center of the round opening in Fig. 1a? You mention in this figure that the 'mask' had a diameter of 10 mm, and you point to the dark area in the center of the round opening and that is not round. At the same time, you say that the non-irradiated areas are the outer (metallic-looking) areas close to the edge of the round opening. Can u please revise Fig. 1a to make clear to the reader what is what? Also make a small technical drawing to explain better what happens in cross-section as well as in top view; the picture of the setup will then be easier to understand.
- In Fig. 1b, you show the damage profile only for the sample with 8 dpa damage on the surface; pls add the profile for the sample with 4 dpa damage on the surface, for a complete overview.
- In Fig. 2, you show the profile of the electromotive force (in V); pls replace this profile with the profile of the LBE dissolved oxygen concentration (Co, jn wt%) - this will be much easier to follow for the reader who is not used to invert in his mind the EMF with the LBE Co.
- In Fig. 2, it would be nice to indicate with arrows the moments of specimen insertion & extraction from the static autoclaves. Insertion normally corresponds to a reduction in LBE oxygen concentration (or an EMF increase), due to the fact that the steel acts as an oxygen getter. Indicating the moments of specimen insertion & withdrawal into the LBE bath is really important, because you mention that (a) the corrosion test starts when the T reaches 350C, and (b) the exposure lasts for 330 hours; however, in Fig. 3a, we see about 50 hrs (probably the 35 hrs you mention in the text before reaching a low LBE oxygen concentration) before the start of the intended 330 h, as well as about 20 hrs after the end of the period of 330 h (probably the time needed to withdraw the sample from the bath - did you open the lid of the autoclave before you started cooling? I do not understand the drop in the EMF; maybe an unforeseen leak?). In other words, please describe with greater accuracy the experiments you performed, allowing the LMC experts to put in better context your findings.
- You have chosen an exposure duration of 330 h, can u please provide a justification to the reader for that choice of yours? Intuitively, I agree with that choice. I have personally performed dissolution corrosion experiments on 316L steels (both solution annealed & cold worked) at 450C, but for tests lasting for more than 1000 h. At 1000 h, the deepest sites ("pits") of LBE dissolution attack were about 10 microns. Hence, I do not expect that the depth of dissolution corrosion after 330 h at 450C will exceed 5 microns (it will probably be closer to 3 microns); this agrees nicely with the depth of radiation damage (Fig. 1b) that does not exceed 3 microns. Irrespective of my opinion and own data (still not published in open literature), it will be nice to justify to the reader your choice of exposure duration.
- You mentioned that after corrosion testing, you used hot oil to remove the LBE residues. Even though this allowed you to perform XRD analysis to determine the type of oxide scale, it was probably not the best possible choice. Especially for steels that have suffered LBE dissolution attack, where LBE is still present in the steel bulk as LBE penetrations that access the steel surface, using a medium that can melt the LBE-affected areas and remove them, bears the risk of altering the

Author Response
Dear reviewer,
Thank you for your clear comments and suggestions from a point of view as liquid metal expert. I’d like to answer to the reviewer’s recommendations as following,
1.
- Ans. English will be revised by native speaker by using MDPI Author Services.
- The description of the irradiation experiment is a bit unclear. Can u please revise Fig. 1a to make clear to the reader what is what?
Ans. I replaced the Fig.1(a) and (b) with some corrections and explanations and revised the sentences concerning the experiments as shown in L91,92,104,106.
- Ans. I explained about the 4 dpa, because the schematic image was inserted in Fig.1(b). L111
- Ans. The figs were replaced from EMF to Co.
- Ans. I revised the Figs and add some sentences as following the reviewer’s comments with greater accuracy as shown in L113,121,125,149.
- Ans. Thank you for supporting this experimental condition and result. Your feeling is intuitively correct. Yes, the 330 h is an ideal corrosion time, which I expect from a view point of radiation damage depth profile, about 2 microns, and corrosion time appreciable to the both cases of low and high oxygen concentration. Some sentences were added from L153 to157.
- Ans. As you point out, the silicone oil usage is a matter of concern because the oil has a high affinity for the corroded surface, that is, easy to penetrate the oxide layer. Then, I don’t use the silicon oil in these days and keep the rare surface condition as much as possible after this experiment. So far vaseline (white petroleum) is used to remove LBE as necessary. I added the sentence showing the risk for the usage of silicone oil in L160.
- Ans. I changed the unit from cps to composition as shown in Fig.5.(b) and (c).
Reviewer 2 Report
Abstract
Line 9 : the term “displacement damage” is understandable to those skilled in the art but is not very clear for use in a summary
Line 19 : « the surface …an iron oxyde » sentence not clear
The term pit is not the correct term for the observed degradation.
One of the main problem with the corrosion of 316L steel in the presence of LBE is the preferential dissolution of nickel and therefore the surface ferritization of this steel.
It is very strange that in the abstract, nickel is not mentioned.
- Introduction
Line 54 and line 56 : Given the very large number of studies that have been carried out, too few references are cited (and not very recent ones).
A summary of the main results regarding the corrosion of 316L steel in the presence of liquid LBE would be welcome in this introduction.
- Experiments
Line 104 : do you know (did you evaluate) the influence of this 10%period immersion in contact of LBE with a important oxygen concentration ?
- Results
Line 136 : do you sure to observe pitting corrosion ? Did you verify that it is not only selective corrosion, for example of the Nickel ? Did you perform analyses of the surface and sub-surface composition (by EDX-MEB or by XPS or Tof-SIMS) ?
Line 142 : in one case, you detected Pb and Bi : did you sure that it is not due to the procedure to remove the Pb and Bi after corrosion tests ?
Line 154 : how with a SEM measure a so small thickness of oxyde layer. With a Tof-SIM : yes it is possible but with a SEM ….How much do you estimate the measurement error?
Lines 155 and 156 and line 163 : do you sure that all the elements of steel could be considered to be dissolved in the LBE. According the reference 13 page 189, with this oxygen concentration and at 450°C, the oxide with chronium could be formed. Could you discuss the results but with this data ?
Figure 4 and figure 5 : Can you put the scale on all the images which would allow you to be sure to consider the correct scale.
Figure 4 a and c : impossible to observe the selective corrosion …
Figure 5 : The EDX-MEB graph is impossible to read
- Discussion
The observed results are not discussed and compared with the corrosion results of the literature : can you add this comparison and this discussion ?
The nickel, a key element in the corrosion resistance of the 316L steel in presence of LBE, is ignored by the authors. Why ?
Author Response
Dear reviewer,
Thank you for your clear comments and suggestions from a point of view as liquid metal expert. I’d like to answer to the reviewer’s comments as following,
Abstract
- Line 9. Ans. I replaced to radiation damage from “displacement damage”.
- Line 19 Ans. The sentence was revised.
- The term pit is not the correct term for the observed degradation. One of the main problem with the corrosion of 316L steel in the presence of LBE is the preferential dissolution of nickel and therefore the surface ferritization of this steel.
Ans. I agree the Ni dissolution and following ferritization is the main issue in case of 316L but they happen crucially at higher temperature above 500C and long corrosion time over 1000 or 3000 hours even at relatively lower temperature like a 450C, in my understanding. It may be the initial state of the ferritization by Ni depletion. I did not check by AFM or other measurements for nothing filled in the weak contrast, then I changed locally corrosion and/or depletion of Ni from “pit” as following your comments as shown in L17 and whole pages.
Introduction
- Line 54 and line 56 : Given the very large number of studies that have been carried out, too few references are cited (and not very recent ones).A summary of the main results regarding the corrosion of 316L steel in the presence of liquid LBE would be welcome in this introduction.
Ans. Sentences on L64-68 were added.
Experiments
- Line 104 : do you know (did you evaluate) the influence of this 10%period immersion in contact of LBE with a important oxygen concentration ?
Ans. I did not evaluate the effects of 10 % period of high Co. But radiation defects is appeared to enhance the chance of oxidization through mutual diffusion of Fe and Oxygen.
Results
- Line 136 : do you sure to observe pitting corrosion ? Did you verify that it is not only selective corrosion, for example of the Nickel ? Did you perform analyses of the surface and subsurface composition (by EDX-MEB or by XPS or Tof-SIMS) ?
Ans. I revised as following your correction.
- Line 142 : in one case, you detected Pb and Bi : did you sure that it is not due to the procedure to remove the Pb and Bi after corrosion tests ?
Ans. I wrote detailed for removing the surface LBE or not removed.
- Line 154 : how with a SEM measure a so small thickness of oxyde layer. With a Tof-SIM : yes it is possible but with a SEM….How much do you estimate the measurement error?
Ans, By using our FE-SEM made by Zeiss we can observe by high magnitude as same as TEM, about 30-50k.
- Lines 155 and 156 and line 163 : do you sure that all the elements of steel could be considered to be dissolved in th LBE. According the reference 13 page 189, with this oxygen concentration and at 450°C, the oxide with chronium could formed. Could you discuss the results but with this data ?
Ans. L225 L238 L284 and L240-243 are added and revised.
- Figure 4 and figure 5 : Can you put the scale on all the images which would allow you to be sure to consider the correct scale
Ans, I put the scale on all the images.
- Figure 4 a and c : impossible to observe the selective corrosion
Ans. I revised as shown already.
- Figure 5 : The EDX-MEB graph is impossible to read
Ans. I revised the graph.
Discussion
- The observed results are not discussed and compared with the corrosion results of the literature : can you add this comparison and this discussion ?
Ans. I added to result and discussion.
- The nickel, a key element in the corrosion resistance of the 316L steel in presence of LBE, is ignored by the authors. Why ?
Ans. As following your comments, I totally agree with you and revised whole paper.